# An online survey of informal caregivers' unmet needs and associated factors

**Alexandra M. J. Denham**[1,2]*, Olivia Wynne[1], Amanda L. Baker[1], Neil J. Spratt[2,3,4], Alyna Turner[1,5], Parker Magin[1], Kerrin Palazzi[6], Billie Bonevski[1]

**1** School of Medicine and Public Health, Faculty of Health and Medicine, University of Newcastle & Hunter Medical Research Institute, Callaghan, NSW, Australia, **2** Priority Research Centre for Stroke and Brain Injury, Hunter Medical Research Institute, New Lambton Heights, NSW, Australia, **3** School of Biomedical Sciences and Pharmacy, University of Newcastle, Callaghan, NSW, Australia, **4** Department of Neurology, Hunter New England Local Health District, John Hunter Hospital, New Lambton Heights, NSW, Australia, **5** Deakin University, IMPACT Strategic Research Centre, School of Medicine, Barwon Health, Geelong, Victoria, Australia, **6** HMRI Clinical Research Design and Statistics (CReDITSS), Hunter Medical Research Institute, New Lambton Heights, NSW, Australia

* Alexandra.Denham@newcastle.edu.au

**Data Availability Statement:** Data are available at the Open Science Framework database (DOI 10. 17605/OSF.IO/F2AEG).

**Funding:** During the development of this work, AMJD was supported by a Research Training

## Abstract

### Purpose/objective

The purpose of this study was to assess the frequency of unmet needs of carers among a convenience sample of carers, and the participant factors associated with unmet needs, to inform the development of interventions that will support a range of caregivers. The aims of this study were to: (1) assess the most frequently reported moderate-high unmet needs of caregivers; and (2) examine the age, gender, condition of the care recipient, and country variables associated with types of unmet needs reported by informal caregivers.

### Research method/design

An online cross-sectional survey among informal caregivers in English-speaking countries was conducted. Self-reported unmet needs were assessed using an unmet needs measure with the following five unmet needs domains: (1) Health information and support for care recipient; (2) Health service management; (3) Communication and relationship; (4) Self-care; and (5) Support services accessibility. Informal caregivers were asked "In the last month, what was your level of need for help with. . .", and the ten highest ranked moderate-high unmet needs presented as ranked proportions. Logistic regression modelling examined the factors associated with types of unmet needs.

### Results

Overall, 457 caregivers were included in the final analysis. Seven of the ten highest ranked unmet needs experienced by caregivers in the last month were in the Self-care domain, including "Reducing stress in your life" (74.1%). Significant associations were found between younger caregiver age (18–45 years) and reporting moderate-high unmet needs in Health Information and support for care recipient, Health service management, and Support services accessibility (all $p$'s = <0.05).

Program (RTP), University of Newcastle PhD Scholarship, https://www.newcastle.edu.au/ and Hunter Medical Research Institute/Emlyn and Jennie Thomas Postgraduate Medical Research Scholarship, https://hmri.org.au/. ALB is supported by an NHMRC Senior Research Fellowship (APP1135901), https://nhmrc.gov.au/. NJS was the recipient of a co-funded National Health and Medical Research Council/National Heart Foundation Career Development/Future Leader Fellowship (APPS1110629/100827), https://nhmrc.gov.au/; https://www.heartfoundation.org.au/. The funders had no role in study design, data collection and analysis, decision to publish, or preparation of the manuscript.

**Competing interests:** The authors have declared that no competing interests exist.

## Conclusions/implications

Caregivers are not experiencing significant differences in unmet needs between countries and caree/care recipient conditions, suggesting that general interventions could be developed to support a range of caregivers across countries. Increased awareness of informal caregivers' unmet needs, particularly for younger caregivers, among health care providers may improve support provision to caregivers.

## Introduction

Globally, individuals with physical disability, a cognitive condition or a chronic life-limiting illness depend on unpaid care provided by family members and friends [1–3]. The official number and percentage of informal carers in the population in each country, released in the 2018 Global State of Care Report [3], in the following high-income countries were as follows: Australia, 2.7 million (11%); Canada, 8.1 million (28%); the United Kingdom (UK), 6.5 million (10.3%); and the United States (USA), 43.5 million (13%). These numbers are expected to increase with an ageing population worldwide. While carers can articulate positives such as pride associated with progress and/or recovery made by the care recipient (also known as the caree) arising from the caring role [4], there is also strong and consistent evidence demonstrating that providing care for others can negatively impact caregivers as the caregivers forego their own health and social well-being to meet the needs of the care recipients[5]. Internationally recognised priorities for caregivers' support include caregivers' needs, awareness, supportive workplaces, and health and well-being [3, 5].

Unmet needs are defined as the differences between the services perceived by the informal caregiver to be necessary to manage both the health condition of the patient and the caregiver, and the services actually received [6, 7]. For informal caregivers of people with health concerns, experiencing unmet needs can result in negative outcomes in mental health, such as isolation and loneliness [8, 9], increased burden, depression, anxiety, and deterioration of physical ill-health, exacerbation of chronic health issues and frailty [5, 8–14]. Previous research conducted on the unmet needs of caregivers of people with cancer have found that the demographic variables of age [15–17] and gender [16–18] are associated with experiencing higher rates of moderate-high unmet needs. In particular, younger caregivers experienced more care-related information, financial and social unmet needs than older caregivers [16]. Female caregivers in this population have also reported higher unmet needs [16–18], particularly unmet psychosocial needs [18], than male caregivers. Therefore, individual impacts can place unique challenges and demands on the caregiver. Further research needs to be conducted to compare these factors and examine their influence on unmet needs of informal caregivers.

It is difficult to compare quality, quantity, utilisation and provision of health care services, even when limited to high-income countries [19, 20], and what this means for informal caregivers who are seeking support. One way to examine these areas of interest is to assess caregivers' unmet needs using the same instrument across countries and care recipient conditions. Comparing informal caregivers' unmet needs assessed by the same instrument across countries and caregiving groups could provide meaningful insight into the unique and common unmet needs of informal caregivers. For example, understanding possible similarities would allow for the efficient sharing of support programs. Alternatively, if the impact of caring is unique to type of health condition or country, tailored programs may need to be developed.

The purpose of this study was to identify the frequency of self-reported unmet needs among a convenience sample of informal carers, and the age, gender, caregiving group and

country-related factors associated with high unmet needs. This information will be used to inform the development of interventions that will target and address frequently reported unmet needs, in order to support a range of caregivers. The aims of this study were to: (1) assess the percentage of participants reporting moderate-high unmet needs across unmet needs domains; and (2) examine the following variables associated with types of unmet needs: demographic (age, gender), care recipient condition (Alcohol and other drug use; Alzheimer's, dementia; Cancer; Mental/emotional illness; Mobility, physical disability; "Old age", frailty; Stroke; and Other), and country (Australia, Canada, New Zealand (NZ), UK, USA and other) variables associated with types of unmet needs reported by informal caregivers.

## Methods

### Study design

An online cross-sectional survey was conducted. Participants were primarily recruited through social media, such as Facebook, Twitter, and relevant online websites, between March and August 2018. The study received approval from the University of Newcastle Human Research Ethics Committee, Approval No. H-2017-0312.

### Setting

This study was conducted via the internet. Participants were targeted by Facebook advertising if they followed caregiving Facebook pages in Australia, Canada, NZ, the UK or the USA. These countries were selected as they were primarily English-speaking high-income countries, but still had sufficient differences in health care systems and countries to meaningfully compare and examine unmet needs across individual demographic and clinical factors. Additional avenues for recruitment included online newsletters, study information posted on relevant websites, forum posts, and clinical registries.

### Sample

Participants were eligible to participate if they reported being: (1) 18 years or older; (2) comfortable using English to participate in the study; and (3) currently an informal caregiver of a person with diminished physical ability or cognitive condition, and/or a chronic life-limiting illness. This included spouses, family members and/or close other individuals such as friends.

### Procedure

Facebook advertising occurred in three ten-day bursts over the recruitment period. In addition to Facebook advertising, various caregivers' organisations were requested to promote the research via their online networks and newsletters. Previous research has found that obtaining informed consent online is not substantially different from obtaining face-to-face consent [21, 22]. Participants were provided an information statement approved by the University of Newcastle Human Research Ethics Committee, in which they were asked to complete an online survey. Participants indicated their consent by completing the 15–20 minute survey.

### Measures

**Sociodemographic characteristics and health conditions.** Demographic information was collected, including country of residence, age (years), gender, marital status, education, employment status, income, main source of income, time spent as a caregiver (years), the relationship between caregiver and care recipient, care recipient age, care recipient gender, whether the caregiver lived with the care recipient, and the chronic health condition

experienced by the care recipient. Care recipient conditions were defined as providing care for someone with: (1) Alcohol or other drug issue; (2) Alzheimer's or other dementia; (3) Cancer; (4) Mental/emotional illness; (5) Mobility, physical disability; (6) "Old age", frailty; (7) Stroke; and (8) Other (please specify).

**Unmet needs measure.** Unmet needs were measured using an unmet needs survey based on the Supportive Care Needs Survey-Partners and Caregivers (SCNS-P&C) [23, 24] that is designed to assess the unmet needs of caregivers to people with cancer (modified items shown in S1 Table). Participants were provided a list of 45 statements describing needs, and they were asked to indicate their level of need for help with the needs statement in the previous month. Participants responded on a five-point scale: (1) *Not Applicable*; (2) *Satisfied*; (3) *Low Need*; (4) *Moderate Need*; and (5) *High Need*. Domain scores (see below for derivation of "domains") were calculated by summing the mean score of each item within the domain. Unmet needs scores of each item within the domains were then: (1) collapsed into "*Moderate or high unmet need*" (score of 4 or 5) vs "*No unmet need*" score of 1,2 or 3); and (2) further collapsed into "*Moderate-high unmet needs*" (reporting at least one moderate-high unmet need response in the domain items) vs "*No unmet needs*" (reporting a non-applicable, satisfied or low need response) for each domain.

In this study, we modified the cancer-specific measure to ensure generalisability and inclusivity across all caregiving groups. The language in the survey was changed to remove cancer-specific items for example: "Accessing information about support services for carers/partners of people with cancer," was modified to, "Accessing information about support services for YOU as a carer/partner". Furthermore, ten additional questions were added which further distinguished unmet needs between the care recipient and the personal unmet needs of the carer, including "Reducing stress in your life", "Helping your care recipient to understand your experience as a carer" and "Addressing fears/concerns about your physical or mental deterioration." To reduce respondent burden, several items which were not shown to be prevalent in previous literature [15, 25, 26] were removed.

Factor analysis was performed using a polychoric correlation matrix (due to inclusion of non-continuous items; Stata Polychoric command), with varimax oblique (oblimin) rotation of factors [27, 28] was performed to explore the factor structure of the modified survey and identify underlying unmet needs domains. The number of factors identified was determined by the eigenvalue <1 rule, in which a single unique variable is indicated, and scree plot [27, 28]. Items were included in the factor where their loadings were the highest [27]. A factor's final composition of items included was also dependent on the clinical relevance of the item based on literature review. Five unmet needs domains emerged from the factor analysis: (1) Health information and support for care recipient; (2) Health service management; (3) Communication and relationship; (4) Self-care; and (5) Support services accessibility. Factor analysis and item loadings are shown in S2 Table. The Kaiser–Meyer–Olkin (KMO) measure [29, 30] was also performed following factor analysis to examine sampling adequacy. The KMO measure ranges from 0.4 (unacceptable) to 0.96 (marvelous), and values of less than 0.60 (mediocre) indicate that sampling is not adequate [29, 30]. The KMO found that the factor analysis sampling adequacy value was 0.78 (middling), indicating that sampling adequacy was acceptable.

## Statistical analyses

Descriptive statistics were presented as count (percent) for participant demographics and compared between countries using Chi-square analysis, or Fisher's exact test where appropriate. To address the first aim, the most frequently reported unmet needs of caregivers were

calculated and reported based on the proportion of caregivers who reported moderate-high unmet needs on each item (either a score of 4 or 5). To address the second aim, logistic regression modelling was performed to examine the crude and adjusted associations between socio-economic, disease-related, and country-related variables associated with reporting moderate-high unmet needs in each domain. The variables included in modelling were decided *a priori* based on previous literature and content knowledge. These variables were age, gender, country, and care recipient condition.

Collinearity of variables was assessed using Variance Inflation Factors (VIFs) to determine the appropriate predictors to be included in modelling. Collinearity was handled by removing one of the collinear predictors from the model, and where this was performed, it was noted. Crude and adjusted odds ratios with 95% confidence intervals (CIs) and p-values were presented for each model. The statistical package Stata v14.0 [31] was used for all statistical analysis. Significance of associations was specified as $p < 0.05$ (2-tailed) *a priori*.

## Results

Overall, 2183 people clicked on the survey, and 591 people entered data into the online survey (Fig 1). Participants were included in the final analysis if they answered at least one question of the unmet needs survey, and as a result 457 responses were included in the analysis. However, not all questions were answered by all respondents and as a result sample sizes differ according to question.

Table 1 presents the demographics of the study sample. More than half of the participants reported providing care for a family member who was not their spouse, such as their parent or child (n = 192, 53.5%). The most commonly reported care recipient condition was "Other Conditions" (n = 121, 26.48%) which included conditions such as Autism (n = 13), intellectual disability (n = 12), and brain injury (n = 12). As only four people indicated that they cared for someone who had an Alcohol or other drug issue, this category was collapsed into the "Other Conditions" category. Countries included in the "Other Countries" country category included Ethiopia (n = 2), Germany (n = 2), and Russia (n = 2).

### Percentage of participants reporting moderate-high unmet need across domains

The ten highest ranked moderate or high unmet needs items reported in the last month by informal caregivers are shown in Table 2. Seven of the ten top reported unmet needs related to the Self-care domain, followed by two items in the Communication and relationship domain, and one in the Health service management domain. A complete list of the items ranked by percentage of moderate-high unmet needs is reported in S3 Table.

Table 3 shows the adjusted self-reported demographic, disease-related and country related variables associated with reporting moderate-high unmet needs in each domain. Caregiver age (years) was significantly associated with reporting moderate-high unmet needs in the three domains: Health Information and support for care recipient; Health service management; and Support services accessibility (all $p < 0.05$). Crude logistic regression modelling is reported in S4 Table.

### Demographic, disease-related, and country-related variables associated with unmet needs

**Health information and support for care recipient unmet needs domain.** Caregivers aged 18–45 reported significantly higher rates of moderate-high unmet needs in this domain compared to caregivers aged 65+ (OR = 3.41, p = 0.012; Table 3).

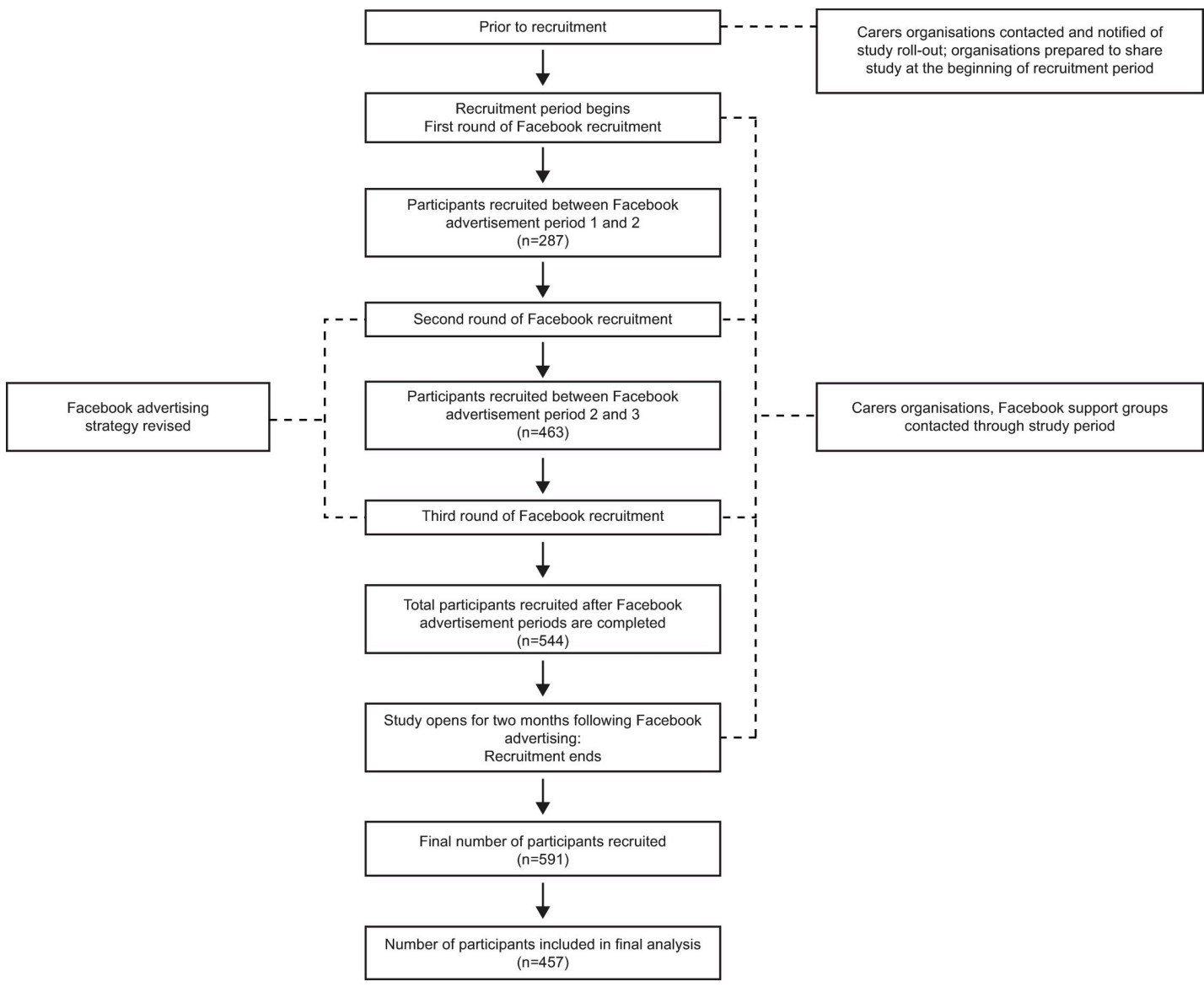

**Fig 1. Recruitment schedule.**

**Health service management unmet needs domain.**   Younger caregiver age was significantly associated with reporting moderate-high unmet needs in the Health service management unmet needs domain ($p = 0.013$), with those aged 18–45 years reporting significantly more moderate-high unmet needs compared to those aged 65+ (OR = 4.24, $p = 0.004$; Table 3).

**Communication and relationship and self-care unmet needs domains.**   Due to the high number of caregivers who reported experiencing moderate-high unmet needs in the Communication and relationships domain (100% reported experiencing at least one moderate-high unmet need) and the Self-care domain (90% reported experiencing at least one moderate-high unmet need), the sample was not powered to perform logistic regression modelling on these domains.

**Support services accessibility unmet needs domain.**   Caregivers aged 18–45 and 45–65 reported significantly higher rates of moderate-high unmet needs in this domain compared to those aged 65+ (OR = 3.76, p = 0.001; and OR = 1.96, p = 0.037, respectively; Table 3).

**Table 1. Demographics of informal caregivers (total n = 457).**

| | n (%) | | | | | | |
|---|---|---|---|---|---|---|---|
| | Overall | Australia | Canada | New Zealand | United Kingdom | United States | Other Countries |
| **Overall sample** | 457 (100) | 202 (44.20) | 67 (14.66) | 43 (9.41) | 42 (9.19) | 91 (19.91) | 12 (2.63) |
| **Caregiver age (years) (n = 457)*** | | | | | | | |
| 18–45 | 109 (23.85) | 47 (23.3) | 9 (13.43) | 11 (25.58) | 8 (19.05) | 27 (29.67) | 7 (58.33) |
| 45–65 | 260 (56.89) | 117 (57.9) | 42 (62.69) | 26 (60.47) | 29 (69.05) | 43 (47.25) | 3 (25.00) |
| 65+ | 88 (19.26) | 38 (18.8) | 16 (23.88) | 6 (13.95) | 5 (11.90) | 21 (23.08) | 2 (16.67) |
| **Gender (n = 457)*** | | | | | | | |
| Male | 62 (13.63) | 32 (15.8) | 4 (5.97) | 2 (4.65) | 5 (11.90) | 16 (17.58) | 4 (33.33) |
| Female | 393 (86.37) | 170 (84.2) | 63 (94.03) | 41 (95.35) | 37 (88.10) | 75 (82.42) | 8 (66.67) |
| **Care recipient condition (n = 457)*** | | | | | | | |
| Alzheimer's, dementia | 84 (18.38) | 28 (13.86) | 12 (17.91) | 10 (23.26) | 12 (28.57) | 18 (19.78) | 4 (33.33) |
| Cancer | 27 (5.91) | 11 (5.45) | 99 (11.94) | 0 (0.00) | 4 (9.52) | 4 (4.40) | 0 (0) |
| Mental/emotional illness | 69 (15.10) | 39 (19.31) | 11 (16.42) | 3 (6.98) | 9 (21.43) | 4 (4.40) | 3 (25.00) |
| Mobility, physical disability | 51 (11.16) | 32 (15.84) | 5 (7.46) | 2 (4.65) | 3 (7.14) | 9 (9.89) | 0 (0) |
| "Old age", frailty | 29 (6.35) | 6 (2.97) | 8 (11.94) | 5 (11.63) | 4 (9.52) | 5 (5.49) | 1 (8.33) |
| Stroke | 76 (16.63) | 26 (12.87) | 11 (16.42) | 7 (16.28) | 3 (7.14) | 27 (29.67) | 2 (16.67) |
| Other Conditions | 121 (26.48) | 60 (29.70) | 12 (17.91) | 16 (37.21) | 7 (16.67) | 24 (26.37) | 2 (16.67) |
| **Education (n = 359)*** | | | | | | | |
| Less than high school education | 66 (18.38) | 41 (24.40) | 5 (9.62) | 7 (21.88) | 7 (23.22) | 5 (7.25) | 1 (12.50) |
| At least high school education | 293 (81.62) | 127 (75.60) | 47 (90.38) | 25 (78.13) | 23 (76.67) | 64 (92.75) | 7 (87.50) |
| **Marital status (n = 359)** | | | | | | | |
| Married | 247 (68.80) | 114 (67.86) | 40 (76.92) | 16 (50.00) | 21 (70.00) | 52 (75.36) | 4 (50.00) |
| Other | 112 (31.20) | 54 (32.14) | 12 (23.08) | 16 (50.00) | 9 (30.00) | 17 (24.64) | 4 (50.00) |
| **Income (n = 359)*** | | | | | | | |
| Low income | 108 (30.08) | 37 (22.02) | 13 (25.00) | 13 (40.63) | 15 (50.00) | 25 (36.23) | 5 (62.50) |
| High/middle | 158 (44.01) | 88 (52.38) | 23 (44.23) | 15 (46.88) | 6 (20.00) | 23 (33.33) | 3 (37.50) |
| Prefer not to answer | 93 (25.91) | 43 (25.60) | 16 (30.77) | 4 (12.50) | 9 (30.00) | 21 (30.43) | 0 (0.00) |
| **Main source of income (n = 358)** | | | | | | | |
| Work (either self-employed, full-time or part-time) | 119 (33.24) | 51 (30.36) | 19 (37.25) | 12 (37.50) | 15 (50.00) | 18 (26.09) | 4 (50.00) |
| Government benefit or pension | 161 (44.97) | 82 (48.81) | 24 (47.06) | 13 (40.63) | 9 (30.00) | 29 (42.03) | 2 (25.00) |
| Family members | 37 (10.34) | 12 (7.14) | 5 (9.80) | 3 (9.38) | 2 (6.67) | 14 (20.29) | 1 (12.50) |
| Personal savings | 40 (11.17) | 21 (12.50) | 2 (3.92) | 2 (6.25) | 2 (6.67) | 7 (10.14) | 1 (12.50) |
| Other | 1 (0.28) | 2 (1.19) | 1 (1.96) | 2 (6.25) | 2 (6.67) | 1 (1.45) | 0 (0) |
| **Employment Status (n = 359)*** | | | | | | | |
| Full time or part time | 121 (33.70) | 54 (32.14) | 19 (36.54) | 11 (34.38) | 16 (53.33) | 17 (24.64) | 4 (50.00) |
| Full-time unpaid caregiver | 124 (34.54) | 72 (42.26) | 8 (15.38) | 16 (50.00) | 4 (13.33) | 24 (34.78) | 1 (12.50) |
| Retired | 85 (23.68) | 31 (18.45) | 18 (36.54) | 4 (12.50) | 6 (20.00) | 23 (33.33) | 2 (25.00) |
| Student | 10 (2.79) | 5 (2.98) | 1 (1.92) | 0 (0) | 1 (3.33) | 2 (2.90) | 1 (12.50) |
| Other | 19 (5.29) | 7 (4.17) | 5 (9.62) | 1 (3.13) | 3 (10.00) | 3 (4.35) | 0 (0) |
| **Years spent caregiving (n = 359)** | | | | | | | |
| Less than 1 year | 24 (6.69) | 5 (2.98) | 4 (7.69) | 4 (12.50) | 2 (6.67) | 8 (11.59) | 1 (12.50) |
| 1–5 years | 138 (38.44) | 60 (35.71) | 16 (30.77) | 12 (37.50) | 15 (50.00) | 31 (44.93) | 4 (50.00) |
| 5–10 years | 74 (20.61) | 36 (21.43) | 14 (26.92) | 7 (21.88) | 5 (16.67) | 9 (13.04) | 3 (37.50) |
| More than 10 years | 123 (34.26) | 67 (39.88) | 18 (34.62) | 9 (28.13) | 8 (26.67) | 21 (30.43) | 0 (0) |
| **Relationship with care recipient (n = 359)*** | | | | | | | |
| Spouse/partner | 153 (42.62) | 69 (41.07) | 28 (53.85) | 8 (25.00) | 8 (26.67) | 38 (55.07) | 2 (25.00) |
| Family member | 192 (53.48) | 91 (54.17) | 24 (46.15) | 23 (71.88) | 22 (73.33) | 27 (39.13) | 5 (62.50) |

(*Continued*)

**Table 1.** (Continued)

| | n (%) | | | | | | |
|---|---|---|---|---|---|---|---|
| | Overall | Australia | Canada | New Zealand | United Kingdom | United States | Other Countries |
| Other | 14 (3.90) | 8 (4.76) | 0 (0) | 1 (3.13) | 0 (0) | 4 (5.80) | 1 (12.50) |
| **Age of care recipient (n = 359)\*** | | | | | | | |
| Younger than 18 | 59 (16.43) | 36 (21.43) | 6 (11.54) | 8 (25.00) | 6 (20.00) | 3 (4.35) | 0 (0) |
| 18–65 | 132 (36.77) | 68 (40.48) | 15 (28.85) | 6 (18.75) | 8 (26.67) | 34 (49.28) | 1 (12.50) |
| 65+ | 168 (46.80) | 64 (38.10) | 31 (59.62) | 18 (56.25) | 16 (53.33) | 32 (46.38) | 7 (87.50) |
| **Care recipient gender (n = 359)** | | | | | | | |
| Male | 202 (56.27) | 89 (52.98) | 38 (73.08) | 14 (43.75) | 17 (56.67) | 39 (56.52) | 6 (62.50) |
| Female | 145 (40.39) | 2 (74.05) | 14 (26.92) | 16 (50.00) | 13 (43.33) | 25 (36.23) | 3 (37.50) |
| Other | 12 (3.34) | 5 (2.98) | 0 (0) | 2 (6.25) | 0 (0) | 5 (7.25) | 0 (0) |
| **Live with care recipient (n = 359)\*** | | | | | | | |
| Yes | 291 (81.06) | 139 (82.74) | 36 (69.23) | 30 (93.75) | 18 (60.00) | 64 (92.75) | 4 (50.00) |
| No | 68 (18.94) | 29 (17.26) | 16 (30.77) | 2 (6.25) | 12 (40.00) | 5 (7.25) | 4 (50.00) |

*Denotes a significant difference (p<0.05) between countries for this characteristic.

Note: Differences in n for each variable are due to missing data.

## Discussion

This paper provides further evidence of the unmet needs of informal caregivers across caregiving groups and countries. Unmet self-care needs were identified as a priority area for support for informal caregivers, in addition to communication and relationship unmet needs. This study also identified that younger caregivers require further support with health information, health services and support services than older caregivers. Caregivers are also experiencing similar unmet needs despite the country and the health condition of the care recipient. Therefore, these data present insights into priority areas for generalisable support to meet the needs of informal caregivers across countries and care recipient conditions.

**Table 2.** Ten highest ranked moderate or high unmet needs reported in the last month by informal caregivers (n = 457).

| Rank | Variable / Item Name | % of sample reporting a moderate-high need | Domain |
|---|---|---|---|
| 1 | Reducing stress in YOUR life (n = 420) | 74.5 | Self-care |
| 2 | Balancing the needs of your caree and YOUR own needs (n = 387) | 71.3 | Communication and relationship |
| 3 | Looking after YOUR own health, including eating and sleeping properly (n = 420) | 71.2 | Self-care |
| 4 | The impact that caring for your caree has had on YOUR working life, or usual activities (n = 402) | 70.6 | Self-care |
| 5 | Taking time off from caregiving (i.e. respite care) (n = 388) | 62.6 | Self-care |
| 6 | Accessing information about support services for YOU as a carer/partner (n = 454) | 59.7 | Self-care |
| 7 | Ensuring there is an ongoing case manager to coordinate services for your CAREE (n = 420) | 54.3 | Health service management |
| 8 | Helping your caree to understand YOUR experience as a carer (n = 387) | 51.2 | Communication and relationship |
| 9 | Accessing information relevant to YOUR needs as a carer/partner (n = 457) | 51.0 | Self-care |
| 10 | Addressing fears/concerns about YOUR physical or mental deterioration (n = 402) | 51.0 | Self-care |

Note: Differences in n for each item are due to missing data.

**Table 3. The adjusted self-reported socioeconomic, disease-related and country related variables associated with reporting moderate-high unmet needs in each domain; logistic regression model (n = 359).**

| | Health Information and Support for Care Recipient | | Health Service Management | | Support Services Accessibility | |
|---|---|---|---|---|---|---|
| | Odds Ratio (CI) | P | Odds Ratio (CI) | p | Odds Ratio (CI) | p |
| Age (years) | | **0.035** | | **0.013** | | **0.005** |
| 18–45 | 3.41 (1.31, 8.86) | **0.012** | 4.24 (1.59, 11.35) | **0.004** | 3.76 (1.68, 8.44) | **0.001** |
| 45–65 | 1.29 (0.66, 2.52) | 0.459 | 1.97 (0.99, 3.92) | 0.052 | 1.96 (1.04, 3.69) | **0.037** |
| 65+ | 1 | | 1 | | 1 | |
| Gender | | | | | | |
| Female | 1 | | 1 | | 1 | |
| Male | 1.3 (0.57, 3.10) | 0.505 | 1.34 (0.56, 3.22) | 0.516 | 1.51 (0.68, 3.36) | 0.307 |
| Country | | 0.199 | | 0.424 | | 0.593 |
| Australia | 1 | | 1 | | 1 | |
| Canada | 1.82 (0.83, 4.04) | 0.139 | 1.74 (0.73, 4.17) | 0.214 | 1.24 (0.59, 2.60) | 0.571 |
| New Zealand | 1.36 (0.50, 3.73) | 0.540 | 0.69 (0.23, 1.95) | 0.482 | 2.00 (0.69, 5.82) | 0.201 |
| United Kingdom | 2.63 (0.91, 7.59) | 0.073 | 1.39 (0.51, 3.80) | 0.523 | 0.88 (0.38, 2.04) | 0.765 |
| United States | 2.02 (0.93, 4.38) | 0.074 | 1.64 (0.72, 3.73) | 0.242 | 0.84 (0.42. 1.67) | 0.615 |
| Caree condition | | 0.107 | | 0.406 | | 0.325 |
| Alzheimer's, Dementia | 1 | | 1 | | 1 | |
| Cancer | 0.84 (0.28, 2.52) | 0.749 | 0.68 (0.22, 2.08) | 0.488 | 1.88 (0.64, 5.57) | 0.253 |
| Mental, emotional illness | 1.85 (0.76, 4.48) | 0.172 | 0.91 (0.38, 2.21) | 0.834 | 1.34 (0.62, 2.92) | 0.454 |
| Mobility, physical disability | 1.18 (0.47, 1.98) | 0.731 | 1.87 (0.62, 5.62) | 0.267 | 0.73 (0.31, 1.69) | 0.457 |
| "Old age", frailty | 0.37 (0.14, 0.99) | 0.048 | 1.26 (0.31, 5.14) | 0.749 | 0.51 (0.18, 1.45) | 0.207 |
| Stroke | 0.80 (0.37, 1.73) | 0.574 | 0.62 (0.28, 1.38) | 0.243 | 0.94 (0.46, 1.90) | 0.855 |

Note: Differences in n for each domain are due to missing data.

Seven of the ten top reported unmet needs were in the Self-care domain, with a further two in the Communication and relationship domain and the last from the Health service management domain. Self-care unmet needs were largely focused on how providing care negatively impacted the caregivers' lifestyle: caregivers struggled to reduce stress, manage their own health, engage in their usual activities/working life, take time off from caregiving, access information about support services to them available as caregivers and information that is relevant to their own unmet needs, and finally, address fears and concerns about their own physical and/or mental deterioration. Opportunities to increase the provision of support in current practice include allowing caregivers to resume usual activities and/or engage in new activities that provide pleasure, respite, and achieve healthier lifestyles that will support caregivers in reducing stress and looking after their own health. Raised awareness of caregiver unmet needs among health care providers should provide direction for support. Recent literature recommends that health providers need competency-based education about the unmet needs of family caregivers [32–36]. Open communication between health care providers and caregivers will assist caregivers to access information relevant to their own unmet needs and help them address fears about their own health deterioration.

The top-rated Communication and relationship unmet needs included balancing the needs of the care recipient with the needs of the caregiver, and helping the care recipient to

understand the experience of the caregiver. As stroke often results in mental and physical outcomes that can make communication challenging, where the care recipient's condition permits, the support of psychological support workers in counselling sessions may be needed to facilitate and assist in healthy discussions of changing roles, perspectives, and understanding of the unique experiences between caregiver and care recipient. Individual support should also be provided to the caregiver. The top-rated Health service management unmet need focuses on the continuation of ongoing support and care for the person with the health condition. While there are few longitudinal studies which investigate the changing needs of caregivers, these studies found that caregivers have high numbers of unmet needs across all domains, which change over time [37, 38]. Previous evidence supports the contention that caregivers experienced difficulty in making arrangements when their care recipient was discharged home, and further difficulties accessing and navigating urgently needed services [39]. As part of their roles, health care professionals such as social workers and case managers should provide ongoing support to caregivers, in addition to the person that they care for, to ensure that they are engaged in much-needed and high-quality care throughout the caregiving continuum and recovery trajectory of their care recipient. Caregivers should be adequately supported and connected with information, services and health plans for themselves and their care recipients with the support of health care professionals to co-ordinate and assist with these tasks.

Younger caregivers (aged 18–45 years) appear to be particularly vulnerable to experiencing unmet needs, as significant differences were shown across the three domains where logistic regression modelling was performed: Health Information and support for care recipient, Health service management, and Support services accessibility. These findings correspond with the existing literature on the unmet needs of caregivers of people who have cancer [15–17]. As the majority of the sample were people who identified as other family members (rather than spouses) who may have taken on the caregiving role to conform to family generational expectations, it is possible that the people who are experiencing the most unmet needs are adult children providing care for an elderly parent, or parents who are providing care for children, a suggestion which is consistent with previous research [40–42]. Further considerations include that carers in this age group are working-age, and may need to manage addition pressures such as reconciling conflicting demands from work and caregiving [43]. This study has identified younger caregiver age as significantly associated with reporting moderate-high unmet needs and may provide health care professionals with additional insight into who may need extra support during the caregiving process to ensure their health and well-being. Further qualitative research should also be conducted to explore the unique unmet needs, challenges and/or coping strategies used by younger family caregivers, with further consideration of differences compared to older family caregivers, spouses and other family members.

Although our study identified that the unmet needs of caregivers were found to be comparable across participant variables, with the exclusion of age, there is a large body of evidence to support the importance of different caregiving scenarios on unmet needs experienced by caregivers. For example, carers of young stroke survivors (less than 65 years old) were found to have higher unmet needs than other carers of stroke survivors [44] and ethnicity may be associated with the long-term unmet needs among carers of stroke survivors [45]. A study investigating the long-term unmet needs of carers of people who have cancer, found that early perceived caregiving stress predicted all domains of unmet needs at the eight year follow-up [46]. Therefore, while the results of this study highlight the frequency of reported unmet needs of younger caregivers, caregivers experiencing many different circumstances are still in need of support to address their unmet needs.

The unmet needs of caregivers were found to be comparable across gender, countries and care recipient conditions. These findings suggest opportunities to increase generalised support

for caregivers across health care systems. The areas this support could address include raising awareness of self-care and communication challenges, in addition to the unique challenges that caregivers continue to face. Future research should consider the development of resources/services to facilitate, enable and support caregivers to effectively communicate their situations and own unmet needs to health care providers, friends, family, and the care recipients so that in turn, they may receive support to meet their needs and increase their quality of life, health, and well-being. To best deliver resources and support to these groups, platforms that have the potential for wide-spread dissemination and reach for provision of resources and support across countries, such as web-based approaches, should be considered.

## Practical implications

These data provide information on the unmet needs of informal caregivers across many caregiving groups and countries, and further provide insights into identifying priority areas of interventions, support and assistance for informal caregivers worldwide. Caregivers are more likely to experience Self-care moderate-high unmet needs, in addition to meeting the unmet needs related to providing care for their care recipients. Coordination among health care providers should be used to acknowledge the concerns of caregivers and encourage them to seek support for their own unmet needs and health. A further consideration arising from this study is that awareness should be raised around the higher risk of unmet needs in younger caregivers aged 18–45 years. Coordination within health care teams such as doctors and nurses should be used to target caregivers to provide them with age-appropriate support, resources and programs.

These results indicate that caregivers across countries and care recipient conditions are experiencing challenges and unmet needs across varying domains, but in particular may be struggling to manage and maintain their own mental and physical health as a result of providing care. Further opportunities include formally recognising the unmet needs of caregivers, in addition to providing them with accessible resources, information and education which consider the unmet needs of the individual caregivers and the complex challenges and impacts of providing care for their care recipients. Furthermore, researchers and service providers should consider the development of programs, resources, and initiatives to support caregivers in customisable and tailored services to monitor and meet their needs. As this sample was recruited online, an online platform/program for delivery of such support may be acceptable and feasible, in addition to the potential for high reach and dissemination capacity to support numerous caregivers across countries and care recipient conditions.

## Limitations

Limitations of this study include some issues around generalisability and representativeness. Firstly, the recruitment process for this study exclusively used online platforms such as Facebook and social media. Therefore, there may be limited representation of both older and younger caregivers who do not regularly use the internet or the targeted platforms may not have had the opportunity to participate in this study. Furthermore, these carers may be more at risk of isolation in need of additional support. However, a strength of using Facebook in this study was the ability to capture a sample of younger caregivers to participate in this research. Furthermore, using these recruitment strategies may identify a sample of participants who are more likely to benefit from interventions and services that have been developed on web-based platforms for online use. One example of an online platform is InformCare [47, 48] developed to support caregivers in the UK. Further issues around generalisability and representativeness include the lack of study materials in languages other than English. Therefore, these results

may not be generalisable to other countries and health care systems, or in lower- and middle-income countries. Further research should focus on exploring the unmet needs of caregivers in these settings and samples.

Another limitation of the current study is the sampling frame. Due to the nature of recruitment using various online platforms, we do not have the sampling frame data to determine how many eligible people received the survey or link; however, the data show that 591 caregivers consented to complete the survey, and responded to some demographic questions. Of those 591, 457 participants completed at least one of the unmet needs questions. There is a growing body of support for Facebook and Twitter as recruitment tools in health research [49, 50]. In particular, previous evidence has supported the use of Twitter for recruiting informal caregivers [50]. Caregivers recruited through Twitter were found to have greater internet proficiency than carers who were recruited using other strategies. Examples of the benefits of using Facebook for study recruitment include better representation, potential of international reach, and increased access to underrepresented and hard-to-reach samples [49–52].

## Conclusions

This study examined the unmet needs of informal caregivers, and the demographic, disease-related and country-related variables associated with reporting moderate-high unmet needs. The highest ranked unmet needs were in the Self-care domain, suggesting that support services and resources for caregivers are needed, particularly those which provide opportunities for caregivers to manage and improve their own health, such as respite care. Additionally, communication and relationship unmet needs are also prevalent; facilitation of healthy discussions between caregiver and care recipient, supported and mediated by health care providers, is needed. Younger caregivers (aged 18–45 years) are particularly at risk for experiencing unmet needs; age-appropriate support and resources should be provided to these caregivers. Country of residence and/or care recipient condition were not directly associated with reported levels of unmet needs in this sample; therefore, generalisable interventions and support may need to be developed. Resources and support disseminated online may be an acceptable and feasible way to support informal caregivers (in particular younger caregivers) across multiple countries who provide care for people with a range of health care conditions.

## Supporting information

**S1 Table. Unmet needs survey.** Modified unmet needs survey based on the Supportive Care Needs Survey-Partners and Caregivers (SCNS-P&C).
(DOCX)

**S2 Table. Factor analysis.** Factor analysis used to establish the types of unmet needs reported by caregivers.
(DOCX)

**S3 Table. Ranked highest moderate or high unmet needs reported in the last month by informal caregivers.**
(DOCX)

**S4 Table. The crude self-reported socioeconomic, disease-related and country related variable associated with reporting moderate-high unmet need in each factor domain; logistic regression model.**
(DOCX)

## Author Contributions

**Conceptualization:** Alexandra M. J. Denham, Olivia Wynne, Amanda L. Baker, Neil J. Spratt, Alyna Turner, Parker Magin, Billie Bonevski.

**Data curation:** Alexandra M. J. Denham.

**Formal analysis:** Alexandra M. J. Denham, Kerrin Palazzi.

**Investigation:** Alexandra M. J. Denham, Olivia Wynne, Amanda L. Baker, Neil J. Spratt, Alyna Turner, Parker Magin, Billie Bonevski.

**Methodology:** Alexandra M. J. Denham, Olivia Wynne, Amanda L. Baker, Neil J. Spratt, Billie Bonevski.

**Project administration:** Alexandra M. J. Denham.

**Resources:** Billie Bonevski.

**Supervision:** Olivia Wynne, Amanda L. Baker, Neil J. Spratt, Billie Bonevski.

**Writing – original draft:** Alexandra M. J. Denham, Olivia Wynne, Amanda L. Baker, Neil J. Spratt, Alyna Turner, Parker Magin, Kerrin Palazzi, Billie Bonevski.

**Writing – review & editing:** Alexandra M. J. Denham, Olivia Wynne, Amanda L. Baker, Neil J. Spratt, Alyna Turner, Parker Magin, Kerrin Palazzi, Billie Bonevski.

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
