## [Decision Letter · Decision Letter 0]

8 Oct 2020

PONE-D-20-26187

­­­­­An online survey of informal caregivers’ unmet needs and associated factors

PLOS ONE

Dear Dr. Denham,

Thank you for submitting your manuscript to PLOS ONE. After careful consideration, we feel that it has merit but does not fully meet PLOS ONE’s publication criteria as it currently stands. Therefore, we invite you to submit a revised version of the manuscript that addresses the points raised during the review process.

We look forward to receiving your revised manuscript.

Kind regards,

Yih-Kuen Jan, PhD

Academic Editor

PLOS ONE

Journal Requirements:

2. Please provide further details on sample size and power calculations.

Reviewers' comments:

Reviewer's Responses to Questions

**Comments to the Author**

1. Is the manuscript technically sound, and do the data support the conclusions?

Reviewer #1: Yes

Reviewer #2: Yes

2. Has the statistical analysis been performed appropriately and rigorously? 

Reviewer #1: I Don't Know

Reviewer #2: Yes

3. Have the authors made all data underlying the findings in their manuscript fully available?

Reviewer #1: Yes

Reviewer #2: No

4. Is the manuscript presented in an intelligible fashion and written in standard English?

Reviewer #1: Yes

Reviewer #2: Yes

5. Review Comments to the Author

Reviewer #1: This is a well-written paper.

I have only a few concerns. The authors did factor analyses, but they did not provide the Kaiser-Meyer-Olkin Measure of Sampling Adequacy estimates. Also, it will be appropriate for the authors to indicate the factor extraction approach and rotation method they employed.

Thank you very much.

Reviewer #2: Thank you for the opportunity to review the manuscript entitled “An online survey of informal caregivers’ unmet needs and associated factors”. The manuscript deals with the important topic of unmet needs for informal caregivers across the caregiving trajectory and highlights the domains of unmet needs at the moderate/high level.

Overall, the manuscript is well-written and the aims, analysis and conclusions are aligned. Below are comments to aid in the clarity of the manuscript.

1) It would be helpful to include the overall purpose for the study that leads into the two aims.

2) The second aim needs to be more clearly articulated.

3) Under the demographics section, line 138, Old Age and Frailty are not synonymous. How were older adults who are not frail delineated?

4) Line 153 – how was the SCNS P&C modified for caregivers of people other than those with cancer?

5) On Table 1 – why does the header indicate Paper 3?

6) The discussion section could provide further details on the importance of distinguishing different caregiving scenarios. For example, there is a significant body of literature that discusses the added strain for caregivers for people living with dementia as well as older adult caregivers who experience a higher level of burden and unmet needs and are themselves often dealing with multiple chronic conditions. The focus on younger caregivers should not be at the exclusion of older adult caregivers.

7) Under the limitations section – the recruitment process may have limited the representation of older adult caregivers who are less reliant upon Facebook and social media and often isolated and in need of additional supports.

6. PLOS authors have the option to publish the peer review history of their article (what does this mean?). If published, this will include your full peer review and any attached files.

Reviewer #1: **Yes: **Pascal Agbadi

Reviewer #2: No

---

## [Author Response · Author response to Decision Letter 0]

3 Nov 2020

Please see Response to reviewer 26-10-2020 that was uploaded with the manuscript documents - Thank you!

19th of October 2020

Yih-Kuen Jan, PhD

Academic Editor

PLOS ONE

Dear Yih-Kuen Jan,

Thank you for your email dated 9th of October 2020 and the opportunity to respond to reviewer comments on Manuscript PONE-D-20-26187 entitled ‘An online survey of informal caregivers’ unmet needs and associated factors'. We thank the editor and the reviewers for their thoughtful comments. Please find below our detailed responses to the reviewers’ comments. We indicate where changes were made and if changes were not made, why we believe they were not possible or appropriate. Additions are highlighted in bolded red during the responses.

 

Journal Requirements

Comment 1:

Thank you – the manuscript has been updated to comply with the style requirements of PLOS ONE.

Comment 2:

Please provide further details on sample size and power calculations.

Page 11, Line 3 - 7

Overall, 2183 people clicked on the survey, and 591 people entered data into the online survey (Fig 1). Participants were included in the final analysis if they answered at least one question of the unmet needs survey, and as a result 457 responses were included in the analysis. However, not all questions were answered by all respondents and as a result sample sizes differ according to question.

Page 13, Line 18-24

Due to the high frequency of caregivers who reported unmet needs in the Communication and relationships domain (100% reported experiencing at least one moderate-high unmet need) and the Self-care domain (90% reported experiencing at least one moderate-high unmet need), the sample was not powered to perform logistic regression modelling on these domains.

Comment 3:

In your Data Availability statement, you have not specified where the minimal data set underlying the results described in your manuscript can be found. PLOS defines a study's minimal data set as the underlying data used to reach the conclusions drawn in the manuscript and any additional data required to replicate the reported study findings in their entirety. 

An online link to the data will be made available https://osf.io/f2aeg/?view_only=ec579bbb77614163868cd6e31974528d using one of the repositories identified on the PLOS ONE page (Open Science Framework).

Comment 4:

Please include captions for your Supporting Information files at the end of your manuscript, and update any in-text citations to match accordingly.

Thank you – the manuscript has been updated to comply with the style requirements of PLOS ONE, including the Supporting Information files - in-text citations have been updated to match accordingly.

 

Reviewer 1

Comment 1: 

The authors did factor analyses, but they did not provide the Kaiser-Meyer-Olkin Measure of Sampling Adequacy estimates. Also, it will be appropriate for the authors to indicate the factor extraction approach and rotation method they employed.

Page 6, Line 14 – Page 7, Line 4

Factor analysis was performed using a polychoric correlation matrix (due to inclusion of non-continuous items; Stata Polychoric command), with varimax oblique (oblimin) rotation of factors [27, 28] was performed to explore the factor structure of the modified survey and identify underlying unmet needs domains. . The number of factors identified was determined by the eigenvalue <1 rule, in which a single unique variable is indicated, and scree plot [27, 28]. Items were included in the factor where their loadings were the highest [27]. A factor’s final composition of items included was also dependent on the clinical relevance of the item based on literature review. Five unmet needs domains emerged from the factor analysis: (1) Health information and support for care recipient; (2) Health service management; (3) Communication and relationship; (4) Self-care; and (5) Support services accessibility. Factor analysis and item loadings are shown in S2 Table. The Kaiser–Meyer–Olkin (KMO) measure [29, 30] was also performed following factor analysis to examine sampling adequacy. The KMO measure ranges from 0.4 (unacceptable) to 0.96 (marvelous), and values of less than 0.60 (mediocre) indicate that sampling is not adequate [29, 30].The KMO found that the factor analysis sampling adequacy value was 0.76 (middling) indicating that sampling adequacy was acceptable.

Reviewer 2

Comment 1: 

It would be helpful to include the overall purpose for the study that leads into the two aims.

Page 1, Line 2

Purpose/objective: The purpose of this study was to assess the frequency of unmet needs of carers among a convenience sample of carers, and the participant factors associated with unmet needs, to inform the development of interventions that will support a range of caregivers. The aims of this study were to…

Page 3, Line 16 - 20

“The purpose of this study was to identify the frequency of self-reported unmet needs among a convenience sample of informal carers, and the age, gender, caregiving group and country-related factors associated with high unmet needs. This information will be used to inform the development of interventions that will target and address frequently reported unmet needs, in order to support a range of caregivers. The aims of this study were to…”

Comment 2:

The second aim needs to be more clearly articulated.

Page 1, Line 6 - 7

The second aim has been updated to read as:

(2) examine the age, gender, condition of the care recipient, and country variables associated with types of unmet needs reported by informal caregivers.

Page 4, Line 1 - 2

(2) examine the following variables associated with types of unmet needs: demographic (age, gender), care recipient condition (Alcohol and other drug use; Alzheimer’s, dementia; Cancer; Mental/emotional illness; Mobility, physical disability; “Old age”, frailty; Stroke; and Other), and country (Australia, Canada, New Zealand (NZ), UK, USA and other) variables associated with types of unmet needs reported by informal caregivers.

Comment 3:

Under the demographics section, line 138, Old Age and Frailty are not synonymous. How were older adults who are not frail delineated?

The conditions categories in this study were used to capture as many short-term physical conditions, long-term physical conditions, emotional/mental health issue and physical conditions as possible. While there is no distinction identified between older adults who are frail and not frail, the category of “Old age”, Frailty was created based on the AARP report on Caregiving in the US (2015) which includes the data of 1,248 carers. In this report, it was found that typically when carers selected no conditions for the care recipient, the recipient’s main problem is reported as “old age” or frailty (National Alliance for Caregiving, 2015). We have also used these categories in our previously published research (Denham, 2019).

National Alliance for Caregiving. "Caregiving in the US 2015." NAC and the AARP Public Institute. Washington DC: Greenwald & Associates (2015). URL: https://www.aarp.org/content/dam/aarp/ppi/2015/caregiving-in-the-united-states-2015-report-revised.pdf

Denham, Alexandra MJ, et al. "An online cross‐sectional survey of the health risk behaviours among informal caregivers." Health promotion journal of Australia: official journal of Australian Association of Health Promotion Professionals 31.3 (2019): 423.

Comment 4:

Line 153 – how was the SCNS P&C modified for caregivers of people other than those with cancer?

Page 6, Line 5 - 8

The language in the survey was changed to remove cancer-specific items for example: “Accessing information about support services for carers/partners of people with cancer,” was modified to, “Accessing information about support services for YOU as a carer/partner”.

Comment 5:

On Table 1 – why does the header indicate Paper 3?

Sincerest apologies – this is a typo, and it has been removed from the manuscript.

Comment 6:

The discussion section could provide further details on the importance of distinguishing different caregiving scenarios. For example, there is a significant body of literature that discusses the added strain for caregivers for people living with dementia as well as older adult caregivers who experience a higher level of burden and unmet needs and are themselves often dealing with multiple chronic conditions. The focus on younger caregivers should not be at the exclusion of older adult caregivers.

Page 16, Line 20 – Page 17, Line 5

Although our study identified that the unmet needs of caregivers were found to be comparable across participant variables, with the exclusion of age, there is a large body of evidence to support the importance of different caregiving scenarios on unmet needs experienced by caregivers. For example, carers of young stroke survivors (less than 65 years old) were found to have higher unmet needs than other carers of stroke survivors [44] and ethnicity may be associated with the long-term unmet needs among carers of stroke survivors [45]. A study investigating the long-term unmet needs of carers of people who have cancer, found that early perceived caregiving stress predicted all domains of unmet needs at the eight year follow-up [46]. Therefore, while the results of this study highlight the frequency of reported unmet needs of younger caregivers, caregivers experiencing many different circumstances are still in need of support to address their unmet needs.

Comment 7:

Under the limitations section – the recruitment process may have limited the representation of older adult caregivers who are less reliant upon Facebook and social media and often isolated and in need of additional supports.

Page 18, Line 19 - 23 

Limitations of this study include some issues around generalisability and representativeness. Firstly, the recruitment process for this study exclusively used online platforms such as Facebook and social media. Therefore, there may be limited representation of both older and younger caregivers who do not regularly use the internet, or the targeted platforms may not have had the opportunity to participate in this study. Furthermore, these carers may be more at risk of isolation in need of additional support. However, a strength of using Facebook in this study was the ability to capture a sample of younger caregivers to participate in this research.

We hope you find this response and associated changes satisfactory. We look forward to publishing in PLOS ONE.

Dr Alexandra Denham

Research Assistant 

School of Medicine and Public Health

Faculty of Health and Medicine

T: +61 2 4033 5712

E: alexandra.denham@newcastle.edu.au

---

## [Decision Letter · Decision Letter 1]

23 Nov 2020

­­­­­An online survey of informal caregivers’ unmet needs and associated factors

PONE-D-20-26187R1

Dear Dr. Denham,

We’re pleased to inform you that your manuscript has been judged scientifically suitable for publication and will be formally accepted for publication once it meets all outstanding technical requirements.

Kind regards,

Yih-Kuen Jan, PhD, University of Illinois at Urbana-Champaign

Academic Editor

PLOS ONE

Additional Editor Comments (optional):

Reviewers' comments:

Reviewer's Responses to Questions

**Comments to the Author**

1. If the authors have adequately addressed your comments raised in a previous round of review and you feel that this manuscript is now acceptable for publication, you may indicate that here to bypass the “Comments to the Author” section, enter your conflict of interest statement in the “Confidential to Editor” section, and submit your "Accept" recommendation.

Reviewer #1: All comments have been addressed

Reviewer #2: All comments have been addressed

2. Is the manuscript technically sound, and do the data support the conclusions?

Reviewer #1: Yes

Reviewer #2: Yes

3. Has the statistical analysis been performed appropriately and rigorously? 

Reviewer #1: Yes

Reviewer #2: Yes

4. Have the authors made all data underlying the findings in their manuscript fully available?

Reviewer #1: Yes

Reviewer #2: Yes

5. Is the manuscript presented in an intelligible fashion and written in standard English?

Reviewer #1: Yes

Reviewer #2: Yes

6. Review Comments to the Author

Reviewer #1: (No Response)

Reviewer #2: (No Response)

7. PLOS authors have the option to publish the peer review history of their article (what does this mean?). If published, this will include your full peer review and any attached files.

Reviewer #1: **Yes: **Pascal Agbadi

Reviewer #2: No

---

## [Editor Report · Acceptance letter]

1 Dec 2020

PONE-D-20-26187R1 

An online survey of informal caregivers’ unmet needs and associated factors 

Dear Dr. Denham:

I'm pleased to inform you that your manuscript has been deemed suitable for publication in PLOS ONE. Congratulations! Your manuscript is now with our production department. 

Kind regards, 

on behalf of

Dr. Yih-Kuen Jan 

Academic Editor

PLOS ONE